# Design and Analysis for Early Warning of Rotor UAV Based on Data-Driven DBN

Xue-Mei Chen [1], Chun-Xue Wu [1,*], Yan Wu [2], Nai-xue Xiong [3], Ren Han [1], Bo-Bo Ju [1] and Sheng Zhang [1]

[1] School of Optical-Electrical and Computer Engineering, University of Shanghai for Science and Technology, Shanghai 200093, China; czrdxm1016@163.com (X.-M.C.); campushr@163.com (R.H.); xh11407130@Outlook.com (B.-B.J.); zhangsheng_usst@aliyun.com (S.Z.)

[2] O'Neill School of Public and Environmental Affairs, Indiana University, Bloomington, IN 47405, USA; yanwu8910@gmail.com

[3] Department of Mathematics and Computer Science, Northeastern State University, Tahlequah, OK 74464, USA; xiongnaixue@gmail.com

* Correspondence: wcx@usst.edu.cn; Tel.: +137-0185-9609

**Abstract:** The unmanned aerial vehicle (UAV), which is a typical multi-sensor closed-loop flight control system, has the properties of multivariable, time-varying, strong coupling, and nonlinearity. Therefore, it is very difficult to obtain an accurate mathematical diagnostic model based on the traditional model-based method; this paper proposes a UAV sensor diagnostic method based on data-driven methods, which greatly improves the reliability of the rotor UAV nonlinear flight control system and achieves early warning. In order to realize the rapid on-line fault detection of the rotor UAV flight system and solve the problems of over-fitting, limited generalization, and long training time in the traditional shallow neural network for sensor fault diagnosis, a comprehensive fault diagnosis method based on deep belief network (DBN) is proposed. Using the DBN to replace the shallow neural network, a large amount of off-line historical sample data obtained from the rotor UAV are trained to obtain the optimal DBN network parameters and complete the on-line intelligent diagnosis to achieve the goal of early warning as possible as quickly. In the end, the two common faults of the UAV sensor, namely the stuck fault and the constant deviation fault, are simulated and compared with the back propagation (BP) neural network model represented by the shallow neural network to verify the effectiveness of the proposed method in the paper.

**Keywords:** rotor UAV; data-driven; on-line; early warning; comprehensive fault diagnosis; DBN

## 1. Introduction

The rotor UAV [1] is an aircraft that does not carry a pilot. It has been widely used in military and civilian fields for its unique advantages, so it is indispensable to ensure the safety and reliability of the rotor UAV flight control system. As an important device for information acquisition, the sensors [2] provide guarantees for the reliable safety of systems. Once faults occur, the flight safety of the rotor UAV will be affected, which will inevitably bring about system control performance degradation. Therefore, rapid detection of sensor failure [3] is a prerequisite for ensuring flight safety.

At present, for rotor UAV flight system sensors, the fault diagnosis method [4] is mostly model based [5], which [6] relies on the accurate model of the system [7]. However, owing to the improvement of computer capabilities, the advancement of artificial intelligence, and ultra-precision technology recently, the rotor UAV flight system has emerged an increasingly complex development trend, which is very difficult to obtain accurate mathematical models [8,9]. In contrast, the data-driven fault

diagnosis method [10] has been proposed due to having no need for obtaining an accurate model of the system. The fault diagnosis can be completed only by using the input and output data of the system. Among them, the neural network method, which has self-association, self-adaptation, and no need to establish accurate mathematical models, are widely used in data-driven fault diagnosis methods. So far, researchers have used neural network methods to conduct extensive research on fault detection technology. The paper [11] presents a method of classifying impact noises obtained from a washer machine by obtaining the time frequency image of the sound signals, which is employed as the input signal to an artificial neural network classifier. A convolutional neural network, which is used to extract the residual signal from different sensor faults into the corresponding time-frequency map and fault characteristics to realize the diagnosis of the UAV sensor, is proposed in the literature [12]. From the visualization, the sensor failure information can be successfully constructed by the convolutional neural network (CNN) extracting the fault diagnosis logic between the residual and the health state. Reference [13] proposed wavelet packet threshold denoising and BP neural network methods for fault diagnosis of rolling bearings. In this method, the Levenberg–Maquardt algorithm was used to improve the traditional BP neural network, which greatly improves the diagnostic level. In reference [14], a novel data-driven adaptive neuron fuzzy inference system (ANFIS)-based approach was proposed to detect on-board navigation sensor faults in UAVs. The main advantages of this algorithm are that it allows the Kalman filter to estimate real-time model-free residual and ANFIS to build a reliable fault detection system. According to the experimental results, it was demonstrated that the method can not only detect fault quickly, but also can be used in real-time applications.

Nevertheless, the traditional shallow neural network method [15] has the disadvantages of over-fitting, local minimum, gradient attenuation, and poor generalization ability, which makes the effect of fault detection unsatisfactory [16].

Therefore, this paper proposes a deep learning method, deep belief network (DBN), instead of shallow neural network. As one of the classic algorithms for deep learning, DBN [17] solves problems such as dimension reduction, information retrieval, and fault classification successfully because of an excellent training algorithm and feature extraction. Therefore, it is applied to the field of fault diagnosis and has certain practicability.

In view of the above discussion, the fault diagnosis of rotor UAV flight control system sensor has been taken as an example and a fault diagnosis method for DBN is presented by this paper. By training a large number of offline historical sample data, the optimal network parameters obtained perform the feature extraction of fault and analyze more essential data features to make it easy to detect faults.

## 2. Mathematical Model of the Rotor UAV Flight Control System

### 2.1. Four-Rotor UAV Model

The Quadrotor AUV [18] is a system controlled by six degrees of freedom with strong coupling, nonlinearity, and interference sensitivity. The four rotors are symmetrically distributed in an "X" shape or a "十" shape, and the center of gravity of the rotor UAV is at geometric center. The power of the UAV is generated by four rotors [19], and the rotation of the rotor produces an upward lift, the magnitude of which is proportional to the square of the angular velocity of the rotor rotation w, that is:

$$F_i = Kw_i^2 \quad i = 1, 2, 3, 4 \tag{1}$$

The Quadrotor AUV controls the attitude and position of the flight through four rotors. The two sets of rotors rotate in the opposite direction to counteract the anti-torsion moment to maintain the attitude stability. The total lift in the vertical direction is generated by four rotors [20], and the rotational speed difference of all the rotors produces the torque of horizontal direction to cause a yawing motion;

the difference in rotational speed between the front and rear rotors controls the pitching motion; the left and right rotors controls the rolling motion [21]. The lift force are expressed as follows:

$$
\begin{aligned}
U_z &= F_1 + F_2 + F_3 + F_4, \\
U_\theta &= bl(-F_2 + F_4), \\
U_\phi &= bl(F_1 - F_3), \\
U_\varphi &= d(-F_1 + F_2 - F_3 + F_4),
\end{aligned}
\tag{2}
$$

where $b, d, l$ are respectively the rotor lift coefficient, the drag index, and the distance from the center of gravity to the axis of the quadrotor UAV; $U_z, U_\theta, U_\phi, U_\varphi$ are respectively the total lift, rolling moment, pitching moment, and yawing moment of the rotor UAV [22]. Through the Newton–Eulerian formula, and assuming that the UAV is in slow flight or hover transition, the kinematics model is obtained. The results are as follows:

$$
\begin{aligned}
\ddot{X} &= (cos\phi sin\theta cos\varphi + sin\phi sin\varphi)U_z/m, \\
\ddot{Y} &= (cos\phi sin\theta cos\varphi + sin\phi sin\varphi)U_z/m, \\
\ddot{Z} &= g - (cos\phi cos\theta)U_z/m, \\
\ddot{\phi} &= \left(\ddot{\theta}\dot{\varphi}(J_Y - J_Z) + U_\theta\right)/J_X, \\
\ddot{\theta} &= \left(\ddot{\phi}\dot{\varphi}(J_Z - J_X) + U_\phi\right)/J_Y, \\
\ddot{\varphi} &= \left(\ddot{\theta}\ddot{\phi}(J_X - J_Y)\right)/J_Z, \dots
\end{aligned}
\tag{3}
$$

where $\ddot{X}, \ddot{Y}, \ddot{Z}$ are the accelerations of the rotor UAV in the ground coordinates and $\theta, \phi, \varphi$ are respectively the roll, pitch, and yaw of the four-rotor UAV. $m$ is the mass of the four-rotor UAV and $g$ is the acceleration of the UAV. $J_X, J_Y, J_Z$ are the moment of inertia of the X shaft, Y shaft, and Z shaft.

## 2.2. Flight Coordinate System Model

### 2.2.1. North East Coast Coordinate System

The Northeast coordinate system [23] is the geodetic coordinate system used by the DJI aircraft. Origin $O_g$ is the take-off point. The three axes of the coordinate system are marked as the right north direction $O_g X_g$, the right east direction $O_g Y_g$, and the vertical ground direction $O_g Z_g$. In the attitude data packet, the north, east, and downward speed curves can be found. In these curves, the value is positive indicating that the speed is north, east, or down.

### 2.2.2. Aircraft Local Coordinate System

The aircraft center point $O_b$ is regarded as the coordinate origin in the aircraft local coordinate system. The three axes correspond to the front and back $O_b X_b$, left and right $O_b Y_b$, and up and down $O_b Z_b$ of the aircraft, respectively; positive and negative apply to the right hand screw rule, as shown in Figure 1.

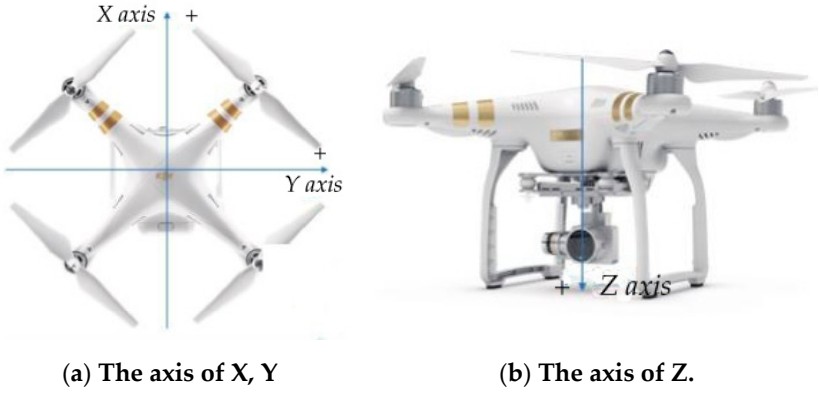

(a) **The axis of X, Y**　　　　　　　　　　　　　(b) **The axis of Z.**

**Figure 1.** *Cont.*

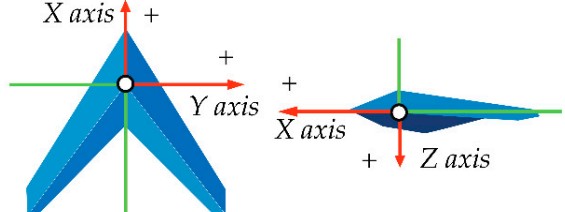

(c) **The definition of Axis direction**

**Figure 1.** Rotor unmanned aerial vehicle (UAV) physical axis in (**a**) the axis of X, Y, (**b**) the axis of Z, and (**c**) the definition of Axis direction.

### 2.2.3. Speed Coordinate Systems

The origin is taken at the center of gravity of the aircraft, and the axis $O_a X_a$ is in the same direction as the flight speed V; the axis $O_a Z_a$ is located on the vertical axis $O_a X_a$ of the plane of symmetry of the aircraft, pointing to the belly; $O_a Y_a$ is perpendicular to the plane $X_a O_a Z_a$, pointing to the right [24]. In the attitude data packet, the north, east, and downward speed curves can be found. In these curves, the value is positive indicating that the speed is north, east, or down.

### 2.2.4. Kinematic Equations for Angular Velocity

In order to describe the movement of the rotor UAV relative to the ground, the geometric relationship between the triaxial attitude angle [25] change rate and the three angular velocity components of the UAV is established as follows:

$$\begin{aligned} \dot{\theta} &= p + qsin\theta tan\phi + cos\theta tan\phi, \\ \dot{\phi} &= qcos\theta - rsin\theta, \\ \dot{\varphi} &= qsin\theta / cos\phi + rcos\theta / cos\phi, \end{aligned} \tag{4}$$

where $p$ represents the rolling rate, $q$ represents the pitching rate, and $r$ represents the yaw rate.

### 2.3. Deep Confidence Network Model

The deep learning idea [26] is inspired by the biological nervous system. It is made up of an input layer, multiple hidden layers, and an output layer. Each layer is connected to each other through nodes or neurons. The output of the previous layer is regarded as the input of each hidden layer. DBN, one of the classical algorithms for deep learning, can automatically extract low-level to high-level, concrete-to-abstract features [27] from raw data through a series of nonlinear transformations and is composed of a number of restricted Boltzmann machines (RBM) [28], which are commonly used to initially set the parameters of the feedforward neural network in order to improve the generalization ability of the model. The RBM network consists of $n$ neurons and $m$ hidden layer neurons. The connection between nodes exists only between layers. RBM comes from the classical thermal theory. The smaller the energy function is, the more stable the system is. The minimum energy of the network is trained to obtain the optimal parameters of the network. The energy function is expressed as follow:

$$E(v,h) = -\sum_{i=1}^{n} a_i v_i - \sum_{j=1}^{m} b_j h_j - \sum_{i,j} v_i h_j w_{ij}, \tag{5}$$

where $v_i$, $h_j$ are respectively the random state of the $i$ unit of the visible layer and the $j$ unit of the hidden layer; $a_i$ and $b_j$ are the corresponding bias; $w_{ij}$ are the weights between the two units. The purpose of training the network is to derive the optimal parameters $(w_{ij}, a_i, b_j)$. The core is the process of using the layer-by-layer greedy learning algorithm to optimize the connection weight of the deep

neural network, which firstly use the unsupervised layer-by-layer training method to effectively mine the fault features in the device to be diagnosed, and then add the corresponding classifier based through the way that reverse supervised fine-tuning to optimize the fault diagnosis capability of DBN. Some nonlinear complex functions can be learned from unsupervised layer-by-layer training by directly mapping data from input to output, which is the key to powerful feature extraction capabilities. Typical network structure [29] is shown in Figure 2.

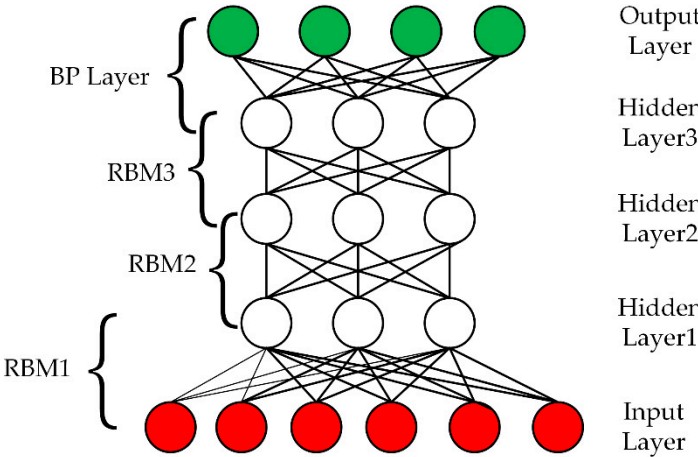

**Figure 2.** Deep belief network (DBN) basic network structure.

## 3. Fault Diagnosis Method Based on Deep Confidence Network

### 3.1. Off-Line Training Based on the DBN Model

#### 3.1.1. Deep Confidence Network Feature Extraction

Deep belief network, a self-learning feature extraction algorithm [30], has been widely used in many application fields with its powerful feature extraction capability [31] and participation without requiring a large amount of tag data. The process of DBN extracting fault features is shown in Figure 3.

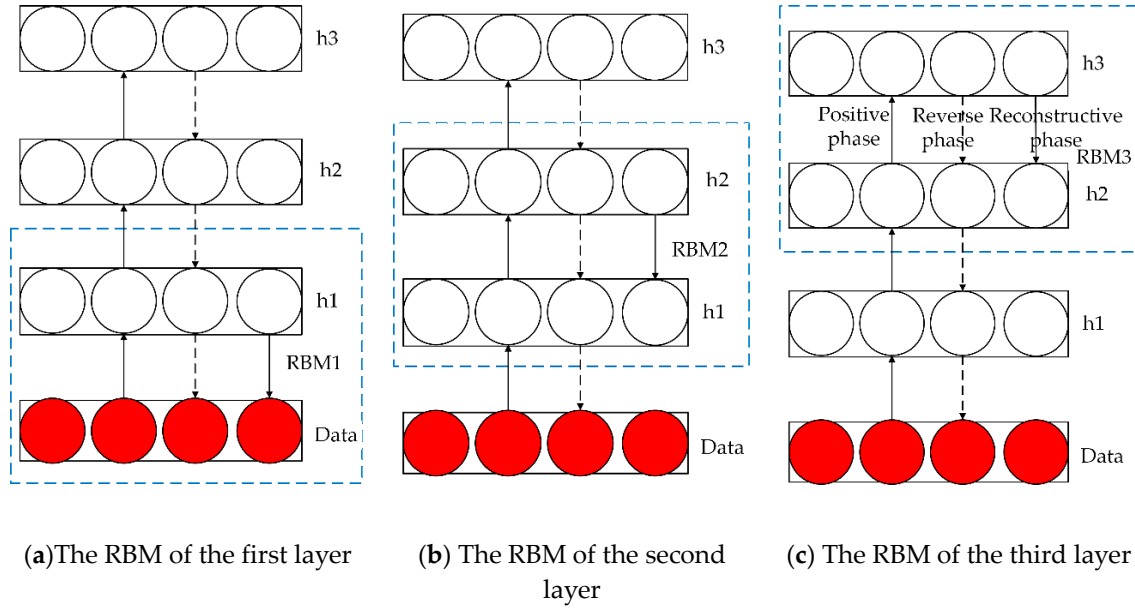

(**a**)The RBM of the first layer　　(**b**) The RBM of the second layer　　(**c**) The RBM of the third layer

**Figure 3.** DBN feature extraction process in (**a**)the restricted Boltzmann machines (RBM) of the first layer, (**b**) the RBM of the second layer, and (**c**) the RBM of the third layer.

The data layer is the visible layer and the initial input data. First, the data vector Data and the first layer hidden layer are used as the first RBM to train the weight $w$ and bias $a$ of the RBM, and then the parameters of the RBM are fixed; h1 is regarded as the visible vector, and h2 is treated as the hidden vector to train the second RBM to get its parameters, namely weight $w$ and bias $b$, and then fix these parameters; finally, to train RBM, which is composed of h1 and h2, the specific training algorithm is shown in Algorithm 1.

---

**Algorithm 1. Description of RBM update algorithm.**

---

**This is the RBM update procedure for binomial units. It can easily adapted to other types of units.**

---

x1 is a sample from the training distribution for the RBM
$\lambda$ is a learning rate for the stochastic gradient descent in Contrastive Divergence
$w$ is the RBM weight matrix, of dimension(number of hidden units, number of inputs)
$a$ is the RBM offset vector for input units
$b$ is the RBM offset vector for hidden units
Notation: $Q(h_2 = 1|x_2)$ is the vector with elements
$Q(h_{2i} = 1|x_2)$
**for all** hidden units $i$ **do**
    compute $Q(h_{1i} = 1|x_1)$ (for binomial units, $\text{sigm}(b_i + \sum_j w_{ij}x_{1j})$)
    sample $h_{1i}\epsilon\{0,1\}$ from $Q(h_{1i}|x_1)$
**end for**
**for all** visible units $j$ **do**
    compute $P(x_{2j} = 1|h_1)$ (for binomial units, $\text{sigm}(a_j + \sum_i w_{ij}h_{1i})$)
    sample $x_{2j}\epsilon\{0,1\}$ from $P(x_{2j} = 1|h_1)$
**end for**
**for all hidden units** $i$ **do**
    compute $Q(h_{2i} = 1|x_2)$ (for binomial units, $\text{sigm}(b_i + \sum_j w_{ij}x_{2j})$)
**end for**

$$w \leftarrow w + \lambda\left(h_1 x_1' - Q(h_2 = 1|x_2)x_2'\right)$$
$$a \leftarrow a + \lambda(x_1 - x_2)$$
$$b = b + \lambda(h_1 - Q(h_2 = 1|x_2))$$

---

### 3.1.2. Deep Confidence Network Training

In fact, the training of RBM is to find the probability distribution that produces the training samples well. Therefore, in order to eliminate the error caused by the data difference between different latitude and longitude, the original data need to be normalized.

$$\begin{aligned} X- &= X.\min(\ ), \\ X+ &= X.\max(\ ). \end{aligned} \tag{6}$$

The DBN learning and training process is mainly divided into two parts:
1. Unsupervised pre-training based on restricted Boltzmann machines from the bottom to the top.

Since the deep belief network is a neural network based on probability model, the decisive factor of its probability distribution depends on the weight $w$, so the goal of the training is to find the best weight. The contrastive divergence (CD) algorithm that finds the best weight is to randomly initialize the parameter set of RBM [32] $w_{ij}, a_i, b_j$. Among them, $a_i$ is the bias of the $i$ node of the visible layer; $b_j$ is the bias of the $j$ node of the hidden layer; $w_{ij}$ is the connection weight of the $i$ node of the visible layer; and the $j$ node of the hidden layer. $\langle\ \rangle_{recon}$ is a reconstructed sample obtained by sampling Gibbs to the sample to estimate the expectation. The learning algorithm is as follows, in Equation (7). Simultaneously, the description of the CD-k algorithm and train unsupervised DBN algorithm are expressed in Algorithms 2 and 3, respectively.

$$\Delta w_{ij} = \lambda\big(\langle v_i h_j\rangle_{data} - \langle v_i h_j\rangle_{recon}\big),$$
$$\Delta a_i = \lambda(\langle v_i\rangle_{data} - \langle v_i\rangle_{recon}),$$
$$\Delta b_j = \lambda\big(\langle h_j\rangle_{data} - \langle h_j\rangle_{recon}\big),$$

(7)

---

**Algorithm 2. CD-k algorithm description.**

---

**Input:** RBM$(V_1, \ldots, V_n, H_1, \ldots, H_m)$, training batch S
**Output:** gradient approximation $\Delta w_{ij}, \Delta a_i$ and $\Delta b_j$ for $i = 1, \ldots, n; j = 1, \ldots, m$.

---

| | |
|---|---|
| 1 | initialize $\Delta w_{ij} = \Delta a_i = \Delta b_j = 0$ for $i = 1, \ldots, n; j = 1, \ldots, m$ |
| 2 | **for** all the $v \epsilon S$ **do** |
| 3 | $v^{(0)} \leftarrow v$ |
| 4 | **for** $t = 0, \ldots, k-1$ **do** |
| 5 |   **for** $i = 1, \ldots, n$ **do sample** $h_i^{(t)} \sim p(h_i|v^{(t)})$ |
| 6 |   **for** $j = 1, \ldots, m$ **do sample** $v_j^{(t+1)} \sim p(v_j|h^{(t)})$ |
| 7 | **for** $i = 1, \ldots, n; j = 1, \ldots, m$ **do** |
| 8 |   $\Delta w_{ij} \leftarrow \Delta w_{ij} + p\big(H_i = 1|v^{(0)}\big)\cdot v_j^0 - p\big(H_i = 1|v^{(k)}\big)\cdot v_j^{(k)}$ |
| 9 |   $\Delta a_i \leftarrow \Delta a_i + v_j^{(0)} - v_j^{(k)}$ |
| 10 |   $\Delta b_j \leftarrow \Delta b_j + p\big(H_i = 1|v^{(0)}\big) - p\big(H_i = 1|v^{(k)}\big)$ |

---

**Algorithm 3. Description of the train unsupervised DBN algorithm.**

---

**Train a DBN in a purely unsupervised way, with the greedy layer-wise procedure in which each added layer is trained as an RBM.**

---

$\hat{P}$ is the input training distribution for the network
$\lambda$ is a learning rate for the RBM training
$\eta$ is the number of layers to train
$w^k$ is the weight matrix for $k$, for $k$ from 1 to $\eta$
$a^k$ is the visible units offset vector for RBM at level $k$, for $k$ from 1 to $\eta$
$b^k$ is the hidden units offset vector for RBM at level $k$, for $k$ from 1 to $\eta$
Mean_field_computation is Boolean that is true if training data at each additional level is obtained by a mean-field approximation instead of stochastic sampling

  **for** $k = 1$ **to** $\eta$ **do**
    **initialize** $w^k = 0, a^k = 0, b^k = 0$
    **while** not stopping criterion **do**
      sample $h^0 = x$ from $\hat{P}$
      **for** $i = 1$ **to** $k-1$ **do**
        **if** mean_field_computation **then**
          assign $h_j^i$ to $Q(h_j^i = 1|h^{i-1})$, for all elements $j$ of $h^i$
        **else**
          sample $h_j^i$ from $Q(h_j^i|h^{i-1})$, for all elements $j$ of $h^i$
        **end if**
      **end for**
      RBMupdate$(h^{k-1}, \lambda, W^k, a^k, b^k)$ {thus providing $Q(h^k|h^{k-1})$ for future use}
    **end while**
  **end for**

---

The CD algorithm is used to train layer by layer for DBN, obtaining the parameters of each layer and initializing the DBN, and then fine-tuning the parameters with the supervised learning algorithm.

2. Supervised tuning training from the top to the bottom.

For supervised tuning training, the forward propagation algorithm is used to obtain a certain output value from the input firstly, and then the backward propagation algorithm is used to update the weights and bias values of the network.

• Forward Propagation Algorithm

1. Pre-trained $w, b$ with the CD algorithm to determine the opening and closing of the corresponding hidden elements. Calculating the stimulus values for each hidden element are as follows:

$$h^{(l)} = w^{(l)} \cdot v + b^{(l)} \tag{8}$$

where $l$ is the layer index of the neural network. The values of $w$ and $b$ are as follows:

$$w = \begin{bmatrix} w_{1,1} & w_{2,1} & \dots & w_{n,1} \\ w_{1,2} & w_{2,2} & \dots & w_{n,2} \\ \dots & \dots & \dots & \dots \\ w_{1,m} & w_{2,m} & \dots & w_{n,m} \end{bmatrix}, b = \begin{bmatrix} b_1 \\ b_2 \\ \dots \\ b_m \end{bmatrix}, \tag{9}$$

where $w_{i,j}$ represents the weight from the $i$ explicit element to the $j$ hidden element.

2. Spread out layer by layer, calculate the excitation value of each hidden element in the hidden layer layer by layer, and standardize it with sigmoid function, as shown below:

$$\sigma\left(h_j\right)^{(l)} = \frac{1}{1 + e^{-h_j}}. \tag{10}$$

3. Finally, the excitation value and output of the output layer are calculated as follows:

$$\begin{aligned} h^{(l)} &= w^{(l)} \cdot h^{(l-1)} + b^{(l)}, \\ \hat{X} &= f\left(h^{(l)}\right), \end{aligned} \tag{11}$$

where $f(\cdot)$ represents the activation function of the output layer and the output value of the output layer is $\hat{X}$.

• Back Propagation Algorithm

1. The error back propagation algorithm of the reconstruction error criterion is used to update the parameters of the whole network and evaluate whether the RBM is trained in the paper. The reconstruction error is the difference between the training data and the original data after the Gibbs sampling by RBM, as shown below:

$$J = \sum_{k=1}^{n} \| v - v^{(k)} \|. \tag{12}$$

The reconstruction error is continuously reduced by iteration times until all RBM training is completed. Finally, the global fine-tuning is performed. Since the last layer of the deep confidence network is used for parameter fitting, the activation function of the last layer selects the hyperbolic tangent function, namely:

$$f(x) = 1 - 1/\left(1 + e^{2x}\right). \tag{13}$$

The output value of the network between −1 and 1 is made. The process of fine-tuning the deep belief network parameters is the process of tuning using the back propagation algorithm. Given the input and output samples, the gradient descent algorithm is used to update the network weights and bias parameters as follows:

$$\left(w^l, b^l\right) \leftarrow \left(w^l, b^l\right) - \lambda \cdot \frac{\partial E}{\partial \left(w^l, b^l\right)}, \tag{14}$$

where $\lambda$ is the learning rate whose numerical value represents the step size of each parameter adjustment. It is generally between 0.005 and 0.200, where $\lambda = 0.10$. In order to overcome the problem that the

training process easily falls into the local minimum value, the impulse term is introduced, and the parameter update direction is inconsistent with the gradient direction. The method is as follows:

$$w_{ij}^{t+1} = mw_{ij}^{t} + \lambda \frac{\partial_\theta}{\partial w_{ij}}, \tag{15}$$

where $m$ is the momentum term, where $m = 0.5$; $t$ is the number of iterations for the sample.

### 3.2. Objective Function Establishment of DBN Fault Diagnosis Model

The deep belief network model is essentially a mapping relationship between input data and output data, that is,

$$
\begin{aligned}
\theta_x &= f\big(H_{x-1}, \dot{H}_{x-1}, W_x, p_x, q_x, \phi_x, \theta_{x-1}\big), \\
\phi_x &= f\big(H_{x-1}, \dot{H}_{x-1}, W_x, q_x, r_x, \theta_x, \phi_{x-1}\big), \\
\varphi_x &= f\big(H_{x-1}, \dot{H}_{x-1}, W_x, q_x, r_x, \theta_x, \varphi_{x-1}\big),
\end{aligned}
\tag{16}
$$

where $H$ stands for the flight height of the aircraft; $\dot{H}$ expresses the rate of change of altitude; $W$ expresses the wind speed.

### 3.3. Online Diagnosis Based on the DBN Model

After the offline training of the deep confidence network is completed, the online estimation can be performed, and the residual between the estimated value and the real value is used to judge whether the fault is faulty. The residual at a certain moment is specifically described as follows:

$$e(t) = \widetilde{o}(t) - o(t). \tag{17}$$

The detection threshold $K_t$ is set for each parameter sensor. By comparing the residual and the threshold, we can determine if there is a fault. When $\left| e(t) \right| < Kt$, it is judged to be faultless; when $\left| e(t) \right| \geq Kt$, it is determined to be faulty. The value of $K_t$ depends on the specific parameters as the case may be. The sensor output value can intuitively determine the stuck fault within a certain period of time, but other sensor fault types such as the constant deviation fault needs to satisfy the mathematical expression of the unknown fault type, as follows:

$$Y_{(t)} = ky_{(t)} + a, t \geq T \tag{18}$$

where $k$ is the failure factor, $a$ is the deviation, and $t$ is the time at which the failure occurred.

Different fault types correspond to different parameters $k$ and $a$ in Equation (18). In the case of a fault within a certain period of time, the DBN estimated value $\widetilde{y}_{(t)}$ is used instead of the true value $y_{(t)}$ in Equation (18), that is:

$$Y_{(t)} = k\widetilde{y}_{(t)} + a, t \geq T. \tag{19}$$

As long as the values of the parameters k and a are known and the estimated values $\widetilde{k}$ and $\widetilde{a}$ are obtained by a function fitting, the category of the fault can be distinguished. The specific fault type and parameter are as follows in Table 1.

**Table 1.** Corresponding of fault type and parameter.

| Fault Type | Parameter |
|---|---|
| Constant deviation fault | $k = 1, |a| \geq K_t$ |
| Constant gain fault | $k \neq 1, |a| < K_t$ |

When the sensor fault is detected by the deep confidence network, the signal reconstruction should be taken in time; that is, the output signal of the sensor is disconnected, and the output of the sensor is replaced by the output estimated by the DBN to ensure that the aircraft continues to fly safely.

## 4. Experiment and Analysis

### 4.1. Experimental Platform

The experiment is run on the Windows Operating System, which is configured as Intel Core i7, 16G memory. The encoding is done on the PyCharm platform that include the TensorFlow Framework. By comparing the model of the traditional BP neural network and the DBN network in the paper, the results will be analyzed accordingly.

The data used in this experiment are derived from the DJI four-rotor UAV. The data in the rotor UAV flight control data record mainly consist of eight parts, namely attitude data, on-screen display (OSD) data, controller data, remote control data, motor data, motor governor data, battery data, and obstacle avoidance data. The data are collected mainly from the attitude data as the source of experimental data in the paper. The attitude data mainly include information on sensors such as position, velocity, angular velocity, accelerometer, gyroscope, magnetometer, barometer, and so on.

The flight data are obtained from the flight attitude data of the rotor UAV. Via the process of normalization, the training samples and the test samples are established. First, the training samples are used for model training, and the weight and bias are continuously adjusted to make it converge quickly, and then the test samples is tested for the fault diagnosis effect to obtain the optimal DBN model. The fault diagnosis process is shown in Figure 4. When collecting data [33], data obtained in various flight state as much as possible, including altitude, altitude change rate, wind speed, pitch, yaw, roll pitch rate, yaw rate, and roll rate. When collecting training data, the flight altitude is 200 m, the flight speed is 40 m/s, the sampling time is 600 s, and the sampling period is 0.1 s.

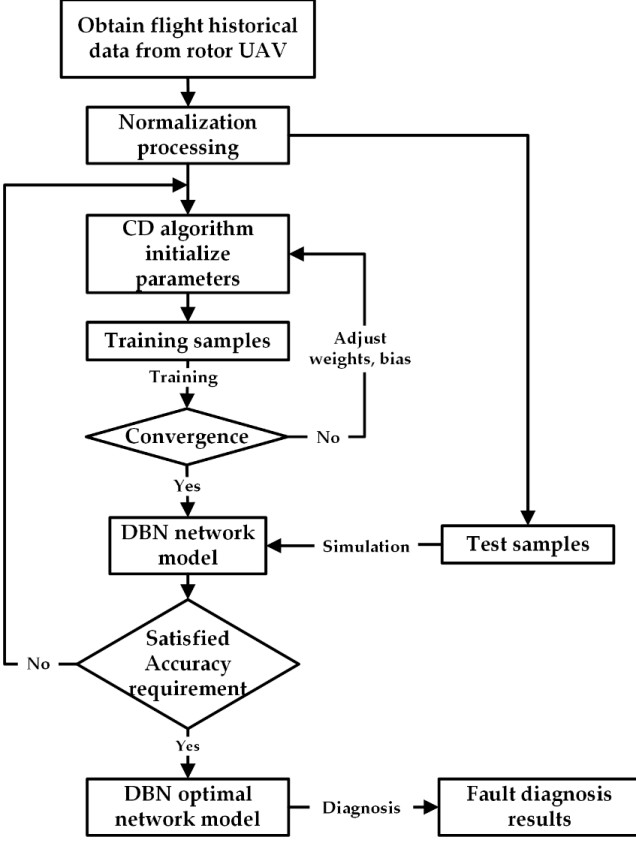

**Figure 4.** Fault diagnosis algorithm flow.

### 4.2. Model Evaluation Index Determination

In order to better analyze the model in the paper, root mean square error (RMSE) and coefficient of determination (R2) are used for the index of regression evaluation. The description is defined as follows.

#### 4.2.1. The Description of the RMSE

RMSE is a measure that reflects the degree of difference between predicted value and actual value. The larger the value is, the larger the difference is. The formula is as follows:

$$\text{RMSE}(y_i, \hat{y}_i) = \sqrt{\frac{1}{n} \sum_{i=1}^{n} (y_i - \hat{y}_i)^2}, \tag{20}$$

where $n$ is the dimension of the sequence, $\hat{y}_i$ represents the prediction value of sensor angular rate, and $y_i$ represents the actual value of sensor angular rate.

#### 4.2.2. The Description of the R2

R2 is also called the goodness of fit. The larger the goodness of fit, the denser the observation point is near the regression line. R2 is in the range between 0 and 1. The larger the value, the better the prediction effect. The expression formula is as follows:

$$R^2(y_i, \hat{y}_i) = 1 - \frac{\sum_{i=1}^{m}(y_i - \hat{y}_i)^2}{\sum_{i=1}^{m}(y_i - \overline{y_i})^2}, \tag{21}$$

where $\overline{y_i}$ is expressed as the average value of sensor angular rate.

### 4.3. Model Structure Selection and Training Results

In the testing, the accuracy of fault diagnosis has a certain relationship with the training samples and the number of RBM layers in the network. When the training samples is different, the RBM layers of the network will also change accordingly. The relationship between the three variables is shown in Figure 5. Firstly, the underlying RBM is constructed with a network model of 3 to 10 layers. The number of neurons in the middle layer is initialized by the random number between 10 and 100. After an overwhelming number of training tests, the iteration times and the reconstruction error convergence curve are as shown in Figure 6.

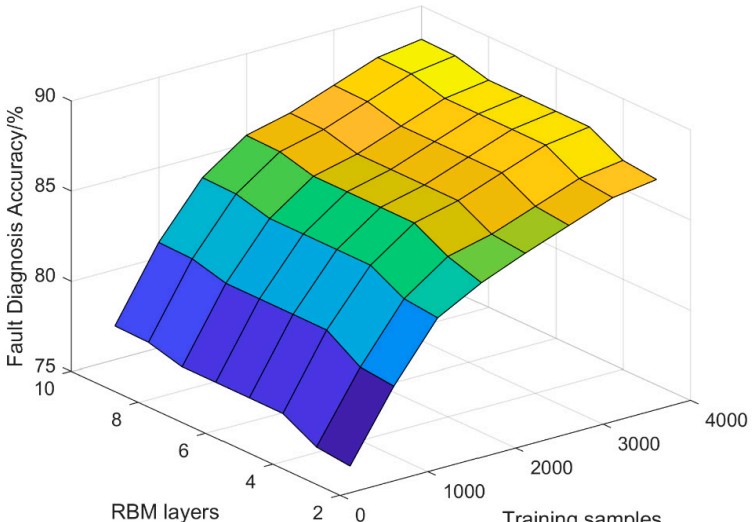

**Figure 5.** Accuracy of different RBM layers and training samples.

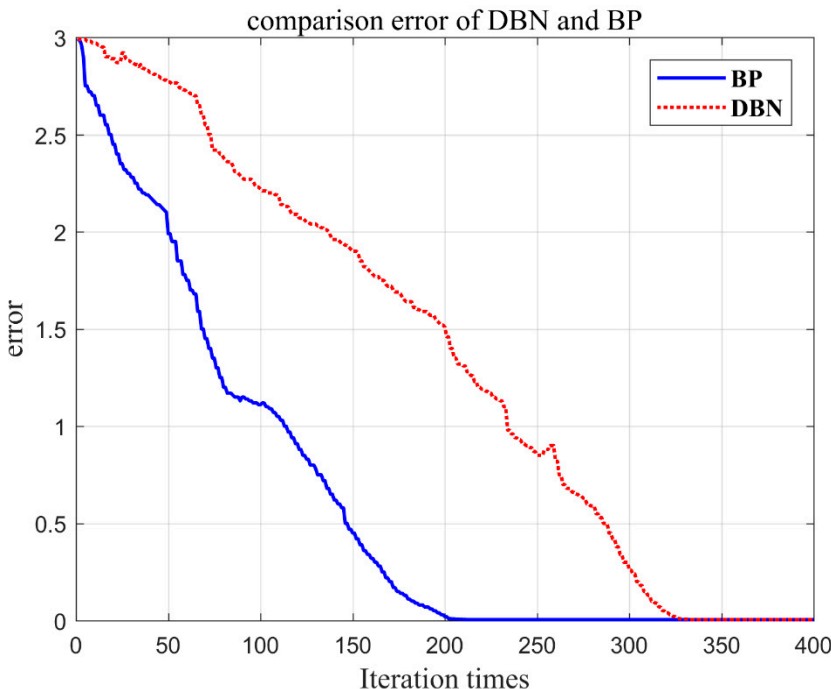

**Figure 6.** The relationship between the number of iteration times and the error.

As can be seen from Figure 6, in the initial stage, as the number of iteration times increases, the reconstruction error decreases rapidly. When the number of iteration time is greater than 200 times, the error reduction gradually stabilizes. Therefore, the number of RBM iteration time per layer is set to 200. On the premise that the training samples is fixed and the number of RBM iterations per layer is set to 200, the diagnostic accuracy of different RBM layers is tested. Change the number of RBM layers, starting at level 0 and modeling only the top level classifier until level 10. The correct relationship diagram is shown in Figure 7.

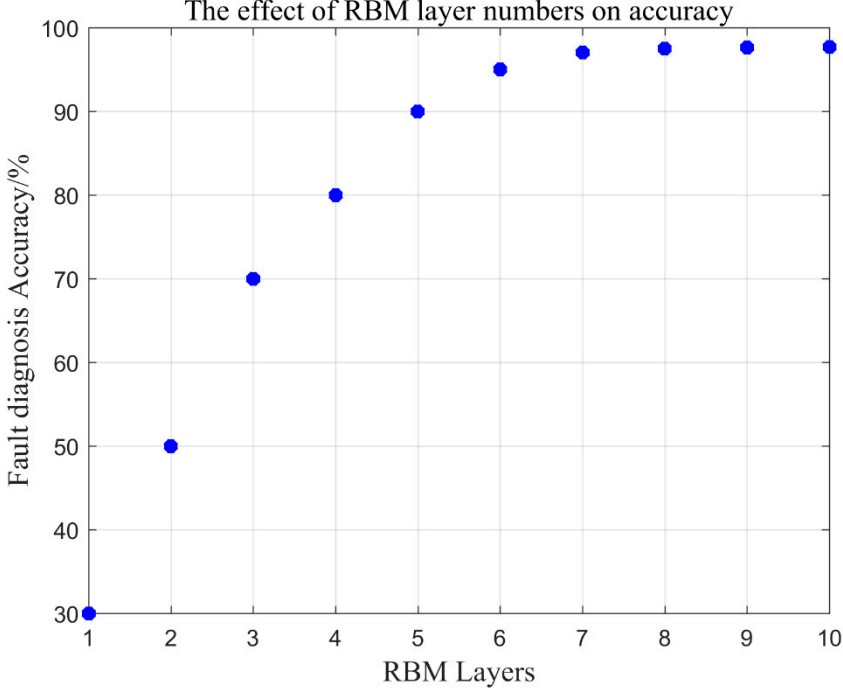

**Figure 7.** Effect of RBM layer number on classification accuracy.

Figure 7 shows that with the increase of RBM layers, the accuracy of fault dignosis is on the rise, and the trend gradually becomes slow. When the RBM is 0 layer, the classification result has the lowest correct rate, only 30%. As the number of layers increases, the discriminative performance of the model increases continuously. However, when the effect of seven layers is reached, the correct rate curve almost no longer rises. The number of bottom RBM layers is six layers and it has reached 95% or more, so the number of RBM layers of the selected model is six layers. Lastly, using the roll sensor, the yaw sensor, and the pitch sensor under normal working conditions, the shallow neural network BP [34] is compared with the deep neural network DBN proposed in the paper, as shown in Figures 8–10.

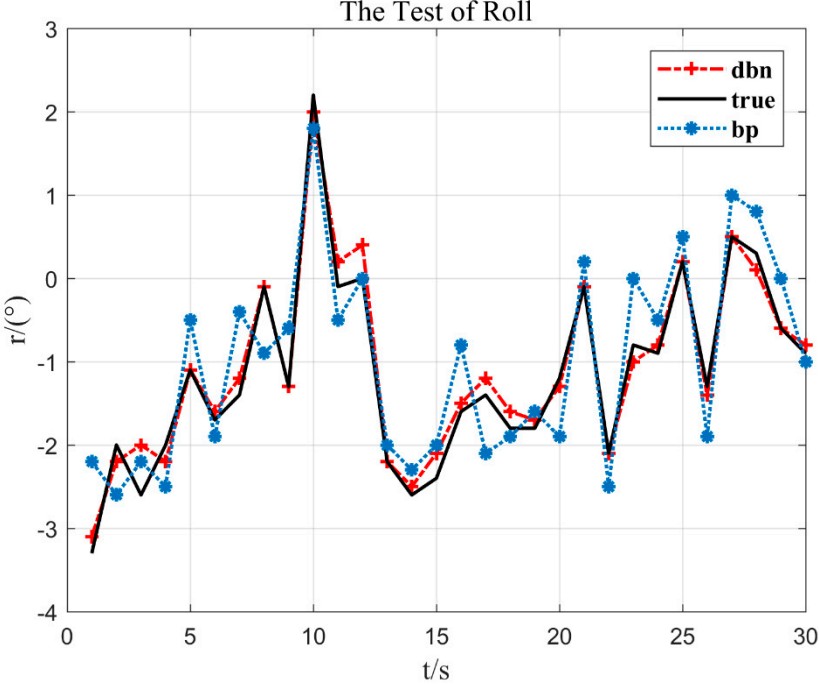

**Figure 8.** The test of roll.

In Figure 8, it can be seen that at 22 s of flight time (x = 22), the flying hand hits about 2% of the crossbar (y = −2.1°) to the left. Then, it leaned to the right again. Corresponding to the actual flight situation, the flying hand went to the left and rolled the bar, but it was released again. According to this situation, the DBN model proposed in this paper can respond well to the stroke situation and has a strong generalization ability.

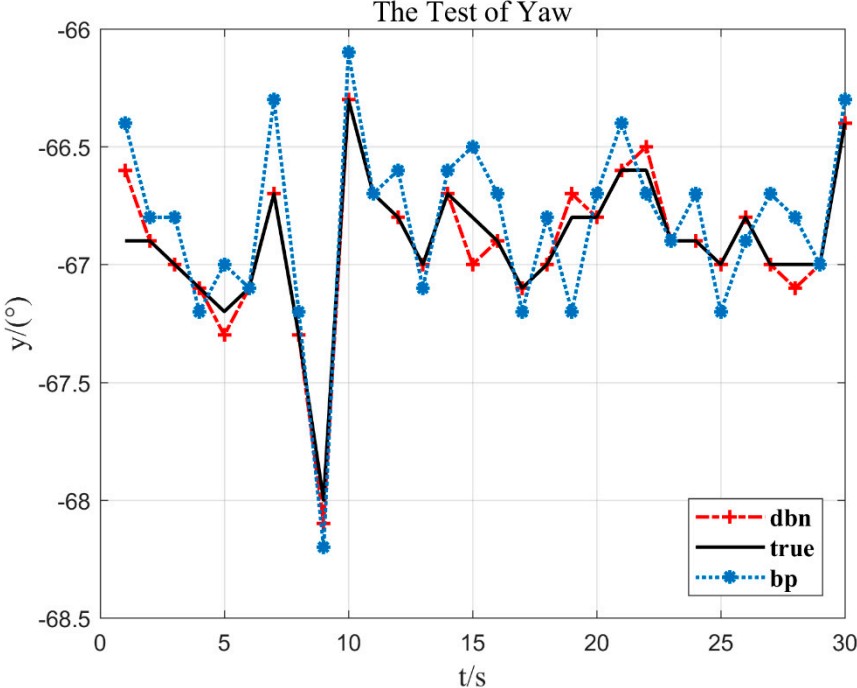

**Figure 9.** The test of yaw.

As can be seen in the trend of the curve in Figure 9, the aircraft yaw is −68° (y = −68°) at 9 s of flight time (x = 9). Nevertheless, at about 10 s, the yaw changes and the value increases from small to large, indicating that the aircraft has rotated clockwise.

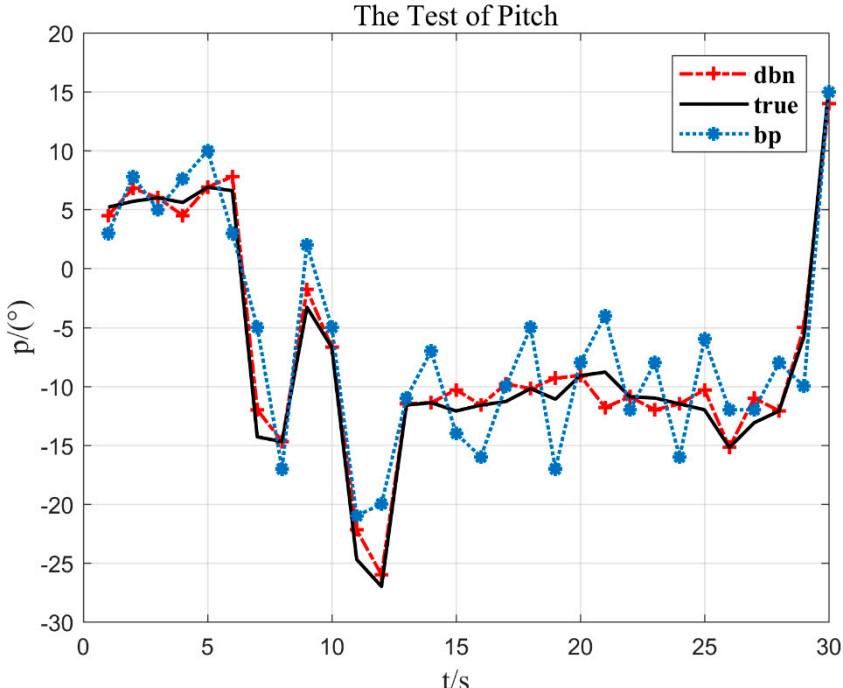

**Figure 10.** The test of pitch.

In Figure 10, the aircraft pitch is stable in the beginning, but between 6 s and 13 s, the aircaft pitch fluctuates violently, which shows that the flying hand went to the left and right continuously. The method DBN proposed in the paper can fit the measured value more accurately.

Table 2 presents the RMSE value and the R2 value between the predicted value and the actual value calculated by the BP neural network and the DBN network of Figures 7–9. As can be seen from Table 2, the RMSE is lower and the R2 is more accurate based on the DBN network. It can be shown that DBN proposed in the paper can better fit the real value of the system compared with the traditional shallow BP neural network, so as to quickly diagnose faults for providing a good foundation.

**Table 2.** Comparison of root mean square error (RMSE) (°)/s and R2 of the two models.

| Models | Roll | Yaw | Pitch | R2 |
|--------|------|-----|-------|-----|
| BP | 10.34 | 156.34 | 74.75 | 0.82 |
| DBN | 1.65 | 89.4 | 25.64 | 0.94 |

*4.4. Experimental Results*

After the training of the DBN model is completed, it can be used for online diagnosis of angle sensor faults. The following is a simulation verification of the pitch, roll, and yaw three sensor in the injection faults.

4.4.1. Sensor Stuck Fault Diagnosis

1. Pitch injection failure

In Figure 11, after the 10 s injection fault, the measurement value does not change any more. The method DBN proposed in the paper can fit the measured value more accurately. When the sensor fault is detected, the output value estimated by the DBN is used to replace the measurement value of the sensor, which provides a guarantee for the safe flight of the UAV.

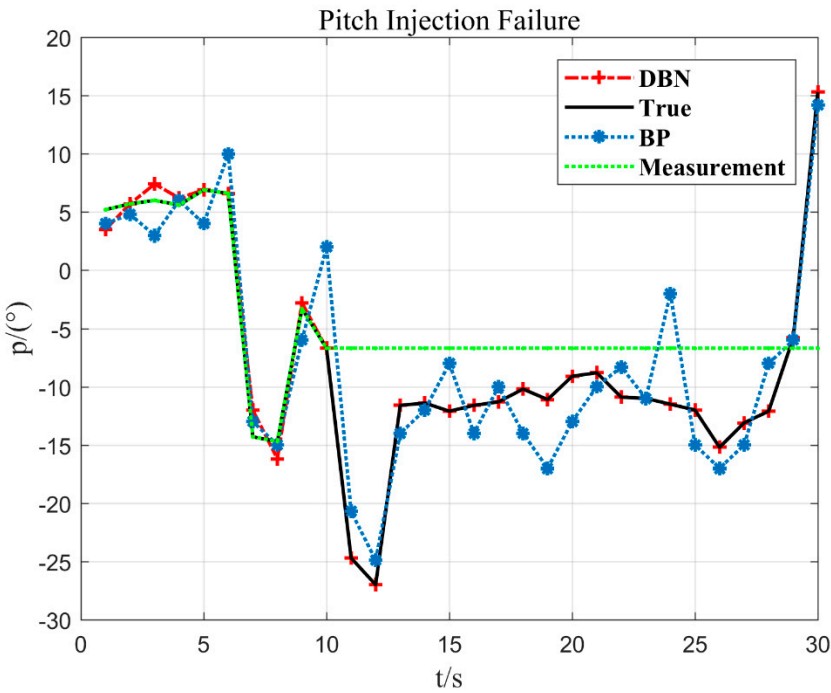

**Figure 11.** Pitch 10 s and −6.7° injection failure.

2. Roll injection failure

In Figure 12, at 10 s of flight time (x = 10), the flying hand hits about 2% of the crossbar (y = 2.2°) to the right, and the duration of the entire crossbar is about 8 s to 10 s. Then, it leaned to the left again. Corresponding to the actual flight situation, the flying hand went to the right and rolled the bar, but it was released again. Therefore, the embodiment of the posture is to tilt to the right and immediately

reverse the brake to slow down. According to this situation, the DBN model proposed in this paper can respond well to the stroke situation. When the sensor fault is detected, the output value estimated by the DBN is used to replace the measurement value of the sensor.

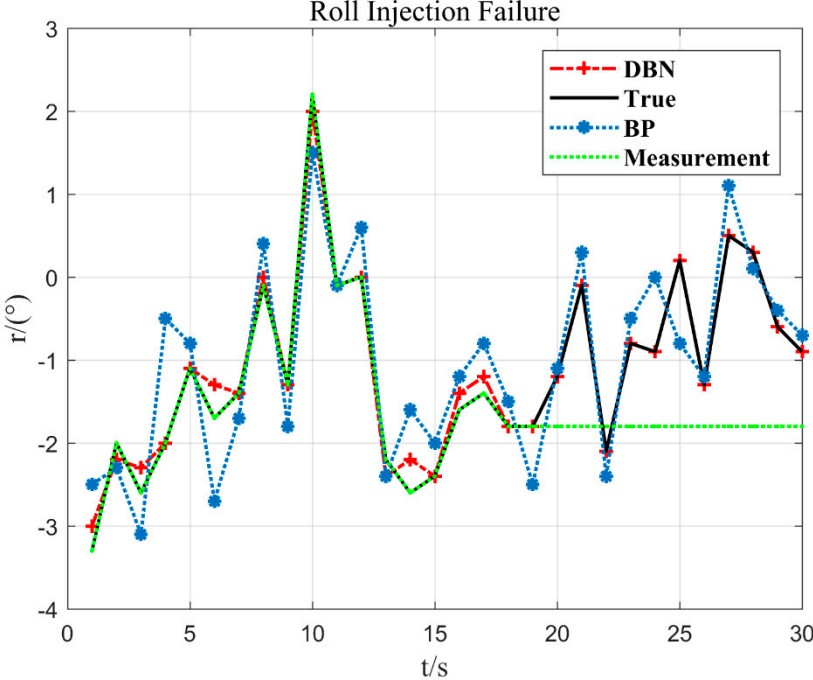

**Figure 12.** Roll 18 s and −1.8° injection failure.

3. Yaw injection Failure

As can be seen in the trend of the curve in Figure 13, the aircraft yaw is −68° (y = −68°) at 9 s of flight time (x = 9). Nevertheless, at about 10 s, the yaw changes and the value increases from small to large, indicating that the aircraft has rotated clockwise. When the sensor fault is detected, the output value estimated by the DBN is used to replace the measurement value of the sensor.

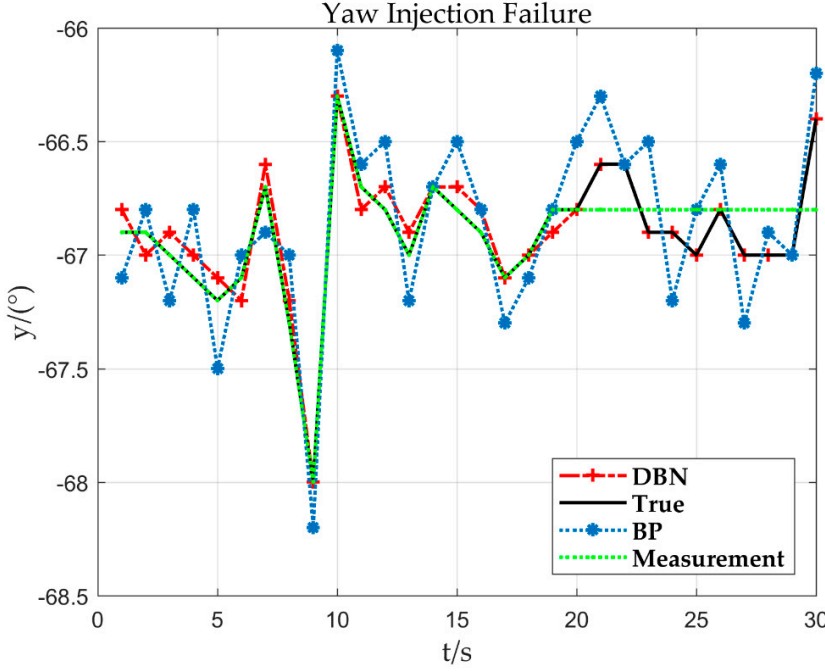

**Figure 13.** Yaw 21 s and −66.6° injection failure.

#### 4.4.2. Sensor Constant Deviation Fault Diagnosis

1. Pitch injection failure

　　Taking the pitch sensor fault as an example, 2°/s constant deviation fault is injected at 10 s, and other simulation conditions are unchanged. The results are as follows.

　　As can be seen from Figure 15, the DBN network has a smaller and more accurate estimation error than the BP network. In order to further judge the type of fault, linearity fitting is used to obtain $\widetilde{k}$ and $\widetilde{a}$. The fault fitting result shown in Figure 14 is $\widetilde{k} = 0.957$, $\widetilde{a} = 2.135$, and the fault corresponding to the deviation is about 2. It can be obtained from Figure 15 that the $K_t$ is set to 1.8. By fitting the parameter $\widetilde{a}$, it can be quickly inferred that the fault type is the sensor constant deviation fault. After the constant deviation fault is identified, the deviation of $\widetilde{a}$ is corrected based on the sensor fault signal to achieve reconstruction of the fault signal.

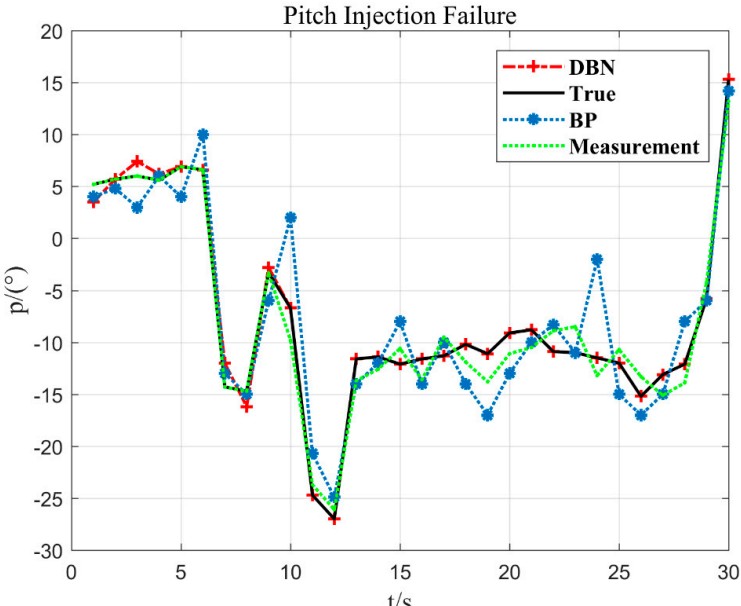

**Figure 14.** Sensor deviation 2°/s pitch response curve.

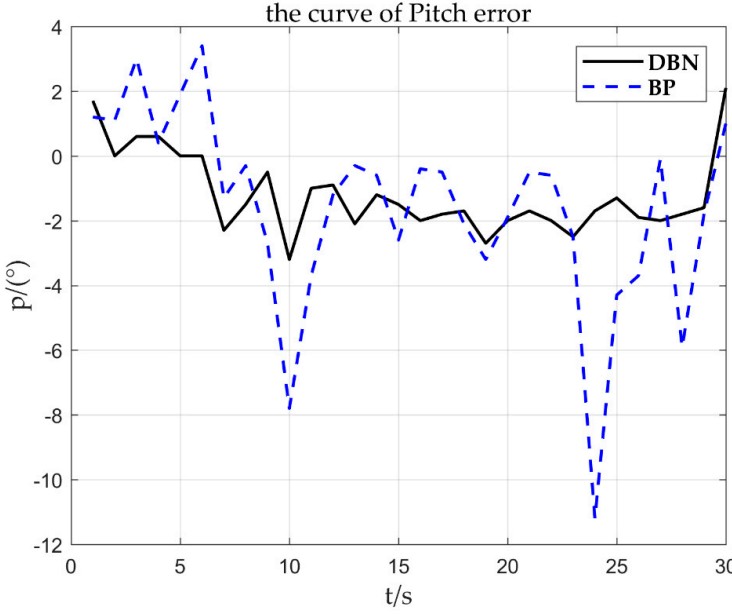

**Figure 15.** 2°/s error curve.

2. Roll injection failure

Taking the roll sensor fault as an example, 0.3°/s constant deviation fault is injected at 18 s, and other simulation conditions are unchanged. The results are as follows.

From Figure 16, the fault fitting result is $\widetilde{k} = 1.023$, $\widetilde{a} = 0.295$, and the corresponding deviation is about 0.3. It can be obtained from Figure 17 that the $K_t$ is set to 0.3. By fitting the parameter $\widetilde{a}$, it can be quickly inferred that the fault type is the sensor constant deviation fault. After the constant deviation fault is identified, the deviation of $\widetilde{a}$ is corrected based on the sensor fault signal to achieve reconstruction of the fault signal.

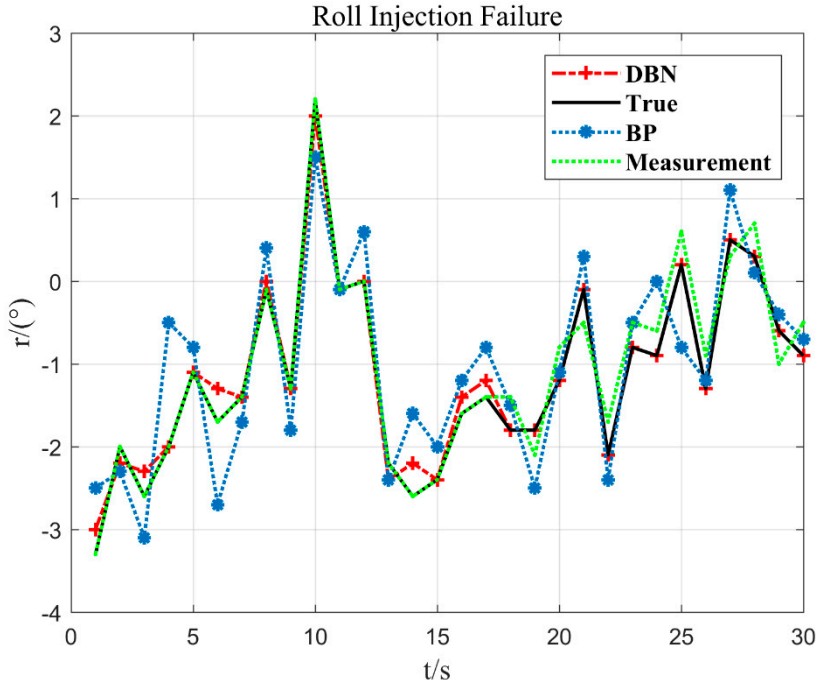

**Figure 16.** Sensor deviation 0.3°/s roll response curve.

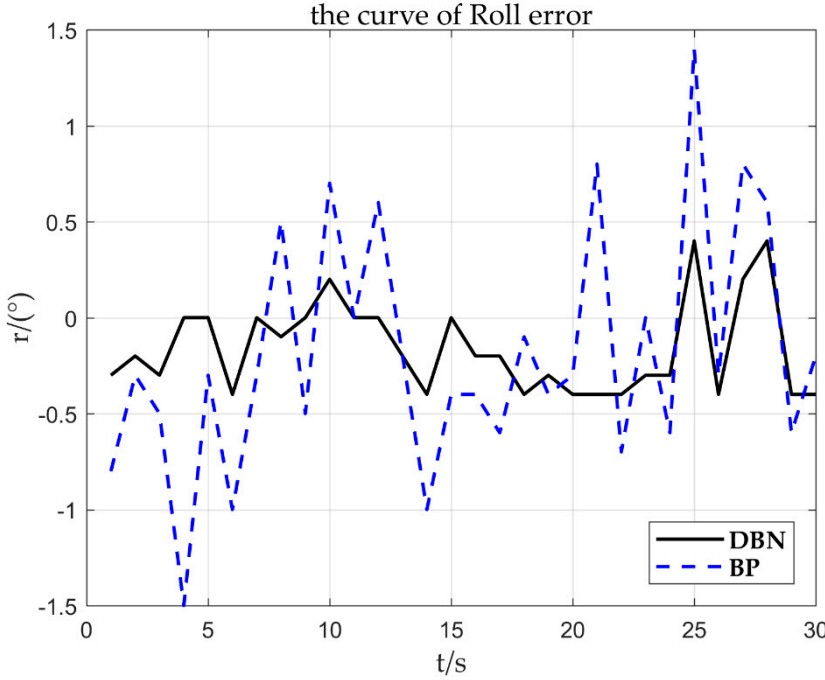

**Figure 17.** 0.3°/s error curve.

3. Yaw injection failure

Taking the yaw sensor fault as an example, 0.08°/s constant deviation fault is injected at 21 s, and other simulation conditions are unchanged. The results are as follows.

The fault fitting result obtained by Figure 18 is $\widetilde{k} = 0.975$, $\widetilde{a} = 0.078$, and the corresponding deviation is about 0.08. It can be obtained from Figure 19 that the $K_t$ is set to 0.07. By fitting the parameter $\widetilde{a}$, it can be quickly inferred that the fault type is the sensor constant deviation fault. After the constant deviation fault is identified, the deviation of $\widetilde{a}$ is corrected based on the sensor fault signal to achieve reconstruction of the fault signal.

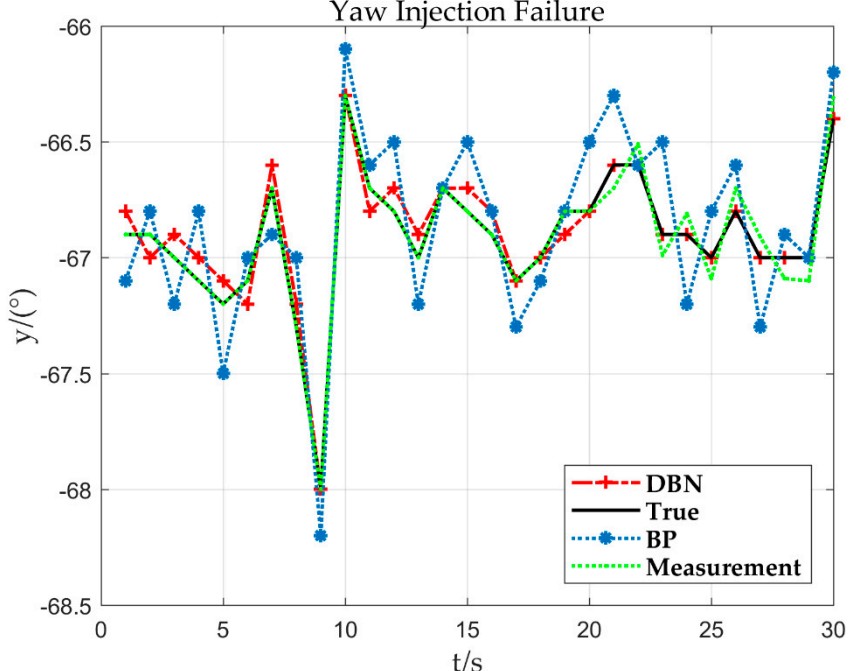

**Figure 18.** Sensor deviation 0.08°/s yaw response curve.

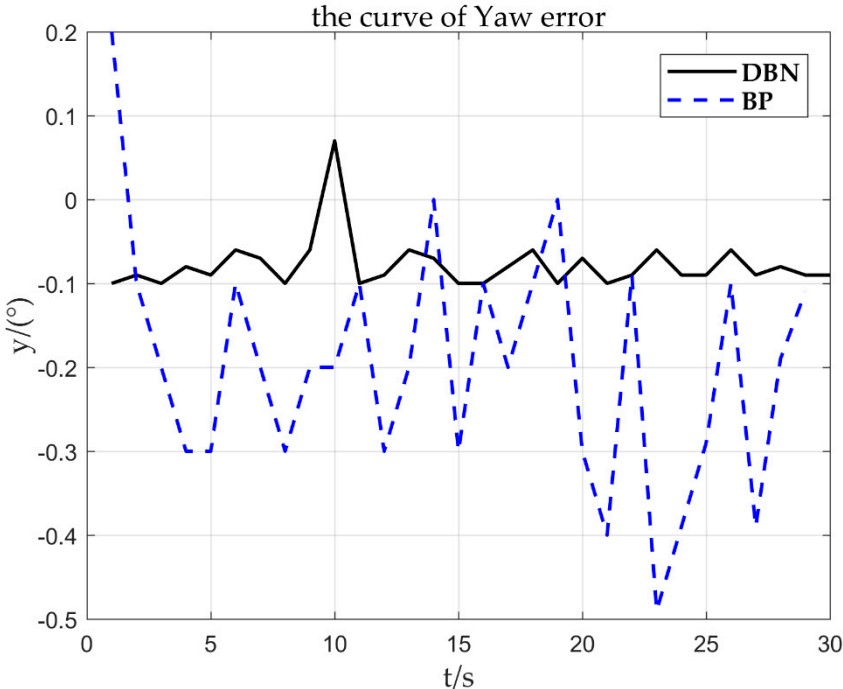

**Figure 19.** 0.08°/s error curve.

From all the above simulation figures, it can be concluded that compared with the traditional BP network model, the DBN network model proposed in the paper can more accurately estimate the UAV's pitch, yaw, roll, and actively respond to the UAV's stroke. Whether the sensor is faulty or not depends on whether the measured value is a fixed value at a certain time. When it comes to faults, fault isolation is immediately performed to ensure safe operation of the rotor UAV flight system.

## 5. Conclusions and Future Works

In the paper, the DBN method based on data-driven methods is applied to the fault diagnosis of the rotor UAV flight system sensors, which effectively solves the problems of the shallow neural networks, such as over-fitting, local minimum, generalization ability, complex functions, insufficient representation ability, and so on, by establishing a deep network structure. The simulation results show that compared with the BP model, the fault diagnosis model has higher convergence speed and diagnostic accuracy and can be extended to sensor diagnosis of other systems. The method currently realizes the real-time fault diagnosis and is applicable to complex the rotor UAV nonlinear systems. On the basis of this, the memory characteristics of long- short-term memory networks (LSTM) can be used to explore the short-term fault prediction of sensors, and to curb the working state of the sensor anytime and anywhere and enable the fault to be effectively prevented before it occurs, which is a necessary means to ensure safe and efficient operation of the rotor UAV. Therefore, the current state of the rotor UAV flight system sensor is estimated in terms of big data mining to realize the system's conditional maintenance and avoid major safety accidents.

In a word, based on the research of the paper, it is indispensable to analyze the different models of different models of UAVs to verify the applicability of the model. In addition, in terms of improving system reliability, it is necessary to carry out deep excavation of the rotor UAV flight data in order to realize the conditional maintenance and life prediction of the equipment.

**Author Contributions:** C.-X.W. and X.-M.C. principally conceived the idea for the study and was responsible for project administration. X.-M.C. and R.H. were responsible for preprocessing all the data and setting up experiments. X.-M.C., Y.W., N.-x.X., B.-B.J., and S.Z. wrote the initial draft of the manuscript and were responsible for revising and improving of the manuscript according to reviewers' comments.

**Funding:** This research was supported by the National Key Research and Development Program of China (No. 2018YFC0810204, 2018YFB17026), National Natural Science Foundation of China (No. 61872242, 61502220), Shanghai Science and Technology Innovation Action Plan Project (17511107203, 16111107502) and Shanghai key lab of modern optical system.

**Acknowledgments:** The authors would like to appreciate all anonymous reviewers for their insightful comments and constructive suggestions to polish this paper in high quality.

**Conflicts of Interest:** The authors declare no conflict of interest. The founding sponsors had no role in the design of the study; in the collection, analyses, or interpretation of data; in the writing of the manuscript, and in the decision to publish the results.

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
