# Peer review of "Design and Analysis for Early Warning of Rotor UAV Based on Data-Driven DBN"

_electronics, doi:10.3390/electronics8111350_

Round 1

Reviewer 1 Report

The paper has been well organized. Here are a few suggestions. 

1) The abstract is missing a motivation statement in the beginning. What is the traditional model-based method ? 

2) Table 2 should be in one page (It is split in two pages)

3) Explain the sub-figures in Figure 7 in more detail. 

4) Is there a reason to iterate to have 10 RBM layers ? What factor effects the number of layers ? 

5) What and how big is the training dataset ? 

Author Response

Dear editor,

        First of all, thank you very much for reviewing my paper and giving me constructive guidance during your busy schedule, which greatly improved the quality of the paper. Based on your comments, the following changes are made.

Point 1:

The abstract is missing a motivation statement in the beginning. What is the traditional model-based method?

Response 1: Please provide your response for Point 1. (in red)

Added the description in the Abstract section to make this discussion more motivated and Explaining the traditional model-based approach is about obtaining accurate mathematical diagnostic models for systems. Traditional methods have advantages for linear systems or simpler systems, but it is difficult to obtain accurate mathematical models for more complex nonlinear UAV systems. So the paper proposes UAV sensor diagnostic method based on data-driven, which greatly improves the reliability of the rotor UAV nonlinear flight control system and achieves early warning (e.g. line 14-17 )

Point 2:

Table 2 should be in one page (It is split in two pages).

Response 2: Please provide your response for Point 2. (in red)

The opinion has been modified in the manuscript. (e.g. line 209)

Point 3:

Explain the sub-figures in Figure 7 in more detail. 

Response 3: Please provide your response for Point 3. (in red)

The sugestion has been supplemented in the paper. ( e.g. line 360-364 - In the Figure 8, it can be seen that at 22 seconds of flight time (x=22), the flying hand hits about 2% of the crossbar (y=-2.1°) to the left. Then leaned to the right again. Corresponding to the actual flight situation, the flying hand went to the left and rolled the bar, but it was released again. According to this situation, the DBN model proposed in this paper can respond well to the stroke situation and hava strong generalization ability; line 367-369 - As can be seen in the trend of the curve in Figure 9, the aircraft yaw is -68° (y = -68°) at 9 seconds of flight time (x=9). Nevertheless, at about 10 seconds, the yaw changes and the value increases from small to large, indicating that the aircraft has rotated clockwise; line 372-374 - In the Figure 10, the aircraft pitch is stable in the beginning, but between 6 seconds and 13 seconds, the aircaft pitch fluctuates violently, which shows that the flying hand went to the left and right continuously. The method DBN proposed in the paper can fit the measured value more accurately.)

Point 4:

Is there a reason to iterate to have 10 RBM layers ? What factor effects the number of layers.

Response 4: Please provide your response for Point 4. (in red)

The 10 layers here are not fixed while change as the training samples change. During the experiment in the manuscript, it was found that there was a certain relationship between the number of RBM layers and the training samples. With the increase of the training samples and the increase of network RBM layers under a certain condition, the fault dignosis accuracy is on the rise, and the trend gradually slowed down. ( e.g. 329-332 - The relationship between the three variables is shown)

Point 5:

What and how big is the training dataset ?

Response 5: Please provide your response for Point 5. (in red)

In the experiment, I selected 6000 pieces of data, 95% of which were used as the training samples and the rest as the testing samples. Mainly obtained altitude, altitude change rate, pitch, yaw, pitch rate, yaw rate, roll rate and wind speed.

        Last but not least, all the modified parts are marked in red in the article. Finally, thank you for taking time form busy to review my article and give me constructive guidance. (Please see the attachment in detail)

Reviewer 2 Report

Mathematical model of a quadrotor motion is written in a chaotic way. Its parts are taken from different papers and does not present of the full model. Assumptions are lacking. First equation in set (2) represents lift force not torque. Coordinates with dots (below equation set (3)) are accelerations and not coordinates. The part of the text in the chapter 2.2.2 describing speed curves should be located in the chapter 2.2.3. So more, the model is not described in the useful form for a control law design. The input, output, and disturbance signals ought be presented in clear form. The same remarks due to diagnostics system used to in the training phase of a neural network. It is interesting how Authors have measured the wind speed during the flight, what was mentioned later. The quadrotor is stabilized by controlled feedback loops. In the loop we have plant, sensors, filters, controller, actuators, etc. The presented method allows to indicate the fault loop. How the Authors are able to  extract the information of the sensor fault from the loop? The quadrotor is not a good plant for such diagnostics system since it has dynamics with small time constants, while the proposed diagnostics system reacts slower. Many of abbreviations are not explained or are too late explained, for example CNN, BP, DBN, OSD. The same variables are described in the text by different symbols, for example flight height is Z and H. Paper should be edited more carefully, see for example captions in Figs 14-16.

Author Response

Dear editor,

        First of all, thank you very much for reviewing my paper and giving me constructive guidance during your busy schedule, which greatly improved the quality of the paper. Based on your comments, the following changes are made.

Point 1:

Mathematical model of a quadrotor motion is written in a chaotic way. Its parts are taken from different papers and does not present of the full model. Assumptions are lacking.

First equation in set (2) represents lift force not torque.

Coordinates with dots (below equation set (3)) are accelerations and not coordinates.

The part of the text in the chapter 2.2.2 describing speed curves should be located in the chapter 2.2.3. 

Response 1: Please provide your response for Point 1. (in red)

The paper is mainly aimed at the analysis of key parameters of rotorcraft under complex disaster conditions. Due to limited data sources, the paper only selects the existing excellent parameters for modeling and analysis. The cited literature is also based on this.

The torque of the rotor UAV is directly proportional to the lift force and the paper has been standardized in terms of description (e.g.line 92).

the description of the equal set (3) is incorrect and has been modified according to your suggestion (e.g.line 109).

the chapter 2.2.2 describing speed curves has been explained has been located in the chapter 2.2.3. (e.g.line 135-136).

Point 2:

So more, the model is not described in the useful form for a control law design. The input, output, and disturbance signals ought be presented in clear form. The same remarks due to diagnostics system used to in the training phase of a neural network. It is interesting how Authors have measured the wind speed during the flight, what was mentioned later.

Response 2: Please provide your response for Point 2. (in red)

In the paper, after analyzing the historical data of various records, the analysis model established by deep learning does not directly introduce the control law model. Regarding the wind speed data, the drone has detection or calculation records during the flight. The flight control data analysis tutorial provides a way to use the attitude ball to reasonably estimate the wind in the high altitude. This method is used for wind acquisition in the manusript. The attitude ball is a visual tool that shows the attitude of the aircraft. Through the blue part of the attitude ball, the attitude change of the aircraft can be seen. In the App interface, click on the small circle in the upper right corner of the "Map" to switch out the "gesture ball". During normal hovering, the blue portion of the attitude ball remains in the spherical lower semicircle. If the blue part is tilted or lifted, it means that the aircraft also has left and right and front and rear attitude changes. In the case where the GPS signal is sufficient and not hit, if the attitude ball changes, it means that the aircraft is resisting the wind and adjusts the posture of the body. The direction in which the posture ball is tilted indicates the direction of the wind. If the aircraft has a continuous horizontal speed change at this time, it means that the current wind speed has exceeded the speed that the aircraft can resist. At this time, the aircraft should be landed or returned as soon as possible.

Point 3:

The quadrotor is stabilized by controlled feedback loops. In the loop we have plant, sensors, filters, controller, actuators, etc. The presented method allows to indicate the fault loop. How the Authors are able to extract the information of the sensor fault from the loop?

Response 3: Please provide your response for Point 3 (in red)

There are many ways to record faults in historical data. The historical data before the fault occurs is the basis of deep learning of the machine. The boundary data of the fault occurs. Some of the faults that do not affect the flight can be judged by the sensor, such as the indicator light failure, etc. Some faults are considered Judging and recording, these different sources of data are used for post-machine learning training. The generated training model is used for flight drones lately and analyzes various parameters for online warning fault diagnosis, which can be used to guide the use of drones by management and maintenance personnel.

Point 4:

The quadrotor is not a good plant for such diagnostics system since it has dynamics with small time constants, while the proposed diagnostics system reacts slower.

Response 4: Please provide your response for Point 4. (in red)

At present, the selection of quadrotor UAV by the research group is based on project needs and relevant regulatory laws, and is also an experimental choice.

Point 5:

Many of abbreviations are not explained or are too late explained, for example CNN, BP, DBN, OSD.

Response 5: Please provide your response for Point 4. (in red)

I have modified the corresponding part of the article (e.g. line 27 - BP neural network is called Back Propagation neural network; Line 55 - CNN is all referred to as Convolutional Neural Network; Line 296 - OSD is called On Screen Display).

Point 6:

The same variables are described in the text by different symbols, for example flight height is Z and H.

Response 6: Please provide your response for Point 4. (in red)

The variables have been standardized in the manuscript. ( e.g. 112 - The uppercase Z is used to indicate the Z-axis; line 263 - H is the variable of flying height. )

Point 7:

Paper should be edited more carefully, see for example captions in Figs 14-16.

Response 7: Please provide your response for Point 4. (in red)

The manuscript has been edited carefully according to the specifications. ( e.g. line 422 - Figure 14. Sensor deviation 2°/s Pitch response curve; line 424 - Figure 15. 2°/s error curve; line 450 - Figure 18. Sensor deviation 0.08°/s Yaw response curve.)

         Last but not least, all the modified parts are marked in red in the article. Finally, thank you for taking time form busy to review my article and give me constructive guidance. (Please see the attachment in detail)

Round 2

Reviewer 2 Report

The response to point 4 in my review is too non-committal.